# The Impact of Lethal, Enforcement-Centred Cat Management on Human Wellbeing: Exploring Lived Experiences of Cat Carers Affected by Cat Culling at the Port of Newcastle

**DOI:** 10.3390/ani13020271

**Published:** 2023-01-12

**Authors:** Rebekah Scotney, Jacquie Rand, Vanessa Rohlf, Andrea Hayward, Pauleen Bennett

**Affiliations:** 1School of Veterinary Science, The University of Queensland, Gatton, QLD 4343, Australia; 2Australian Pet Welfare Foundation, Kenmore, QLD 4069, Australia; 3School of Psychology and Public Health, La Trobe University, P.O. Box 199, Bendigo, VIC 3552, Australia

**Keywords:** free-roaming cats, animal caregiver stress, traumatic stress, cat cull impacts

## Abstract

**Simple Summary:**

Free-roaming cats in urban areas frequently cause complaints. In Australia, cats are classed as domestic or feral depending on how and where they live, with cat management practices varying depending on the cats’ classification. Cats classified as feral can be managed, when considered appropriate by authorities, by shooting them. In 2020, this approach was employed to manage urban cats being fed daily by cat caregivers. This qualitative study aimed to document the lived experience of these cat caregivers to understand their motivations for caregiving and their relationships with these cats. A secondary aim was to explore caregiver perceptions of the lethal management approach and if psychological impacts were experienced. Several main themes arose from interviews with caregivers. The results demonstrate strong relationships between the caregivers and the cats, and negative impacts on caregiver mental health and quality of life associated with this lethal cat management practice. It is recommended that a care-centred approach be taken, whereby authorities identify and assist caregivers to implement neutering and, if possible, adoption. This would improve cat welfare, minimize public complaints, and reduce psychological hazards to caregivers. Legislative amendments should be prioritized to facilitate these recommendations and a revision of the classification between domestic and feral cats should be actioned.

**Abstract:**

In urban and peri-urban areas of the world, free-roaming cats often pose management challenges for authorities. Most are wandering owned or semi-owned cats (fed by people who do not perceive ownership). Some are lost or abandoned, or unowned cats who obtain food from humans unintentionally. Unidentified cats are classified as “stray” in shelter data, and by government agencies as “stray” or “feral” based on their behaviour. However, legally feral cats are usually considered to live and reproduce in the wild with no support from humans. Cats classified as feral in Australia can be managed using lethal methods, including shooting, poisoning, trapping, and blunt trauma. The impact of killing animals on shelter staff is well documented. However, no previous research has investigated psychological impacts of lethal cat management on citizens who care for free-roaming cats. Using semi-structured interviews, this study explored the lived experience of six cat caregivers affected by lethal management of cats by shooting, instigated by the Port of Newcastle in 2020. Results demonstrated strong relationships between the caregivers and cats, and negative impacts on caregiver psychological health and quality of life associated with lethal management. It is recommended that a care-centred approach to cat management be prioritized in future, whereby authorities aid neutering and, if possible, adoption, to improve cat welfare, minimize cat nuisance complaints, and reduce psychological hazards to caregivers. Further, a revision of relevant legislation used to distinguish between domestic and feral cats in Australia should be actioned to prevent unnecessary killing of domestic cats.

## 1. Introduction

In urban and peri-urban areas of the world, free-roaming cats often pose a management challenge for authorities [1]. Management is important because there are concerns about free-roaming cats’ negative environmental impacts. These include the effects of nuisance behaviours [2], such as urinating, defecating and fighting; perceived risk of disease spread to humans, pets, and wildlife; and wildlife predation [3,4]. In Australia there is heightened concern regarding free-roaming cats because of evidence feral cats are a contributing factor to the extinction of native animals, and because of reports estimating that large numbers of native animals are caught by cats in urban and peri-urban areas [4,5,6,7,8,9]. There are also concerns for the health, welfare, and safety of free-roaming cats [2]. Unfortunately, existing management approaches for free-roaming cats typically have not achieved any long-term decrease in the number of cat-related complaints, or the number of cats subsequently impounded by authorities [10,11,12,13,14]. Hence, cat management remains an ongoing issue in many municipalities.

Under government legislation relating to domestic animal management, biosecurity, and feral pests, cats in Australia are considered either domestic or feral. While these terms are inadequately defined in legislation [15], the classification is important, because cat management, prescribed under various state government acts, reflects the terminology used. In New South Wales (NSW), for example, under the Companion Animal Act 1998 [16], cats that are companion animals (domestic cats) are required to be identified with a collar and tag or microchipped by 12 weeks of age, and to be registered (licensed) by 6 months of age. The act relates to companion animals (dogs and cats) but also states “the fact that an animal is not strictly a “companion” does not prevent it being a companion animal for the purposes of this Act”.

In NSW, cats considered companion animals are prohibited from food preparation and consumption areas and designated wildlife protection areas, but are allowed to roam off their owners’ property provided they do not cause a nuisance [16]. If free-roaming domestic cats result in complaints to local government authorities, methods such as trapping are often employed as a management strategy. Typically, complainants are loaned a trap cage and deliver the trapped cat to the local government animal management facility. Owners are contacted to reclaim trapped cats identified through collar tags or microchip databases. If no form of identification is present, the cat is held for a mandated period of typically between 3 and 7 days depending on the state, after which any cat not reclaimed by owners can be either rehomed or killed.

In Australia, feral cats [17,18] are considered to live and reproduce in the wild and survive by hunting and scavenging, with none of their needs met by humans [19]. Feral cats are regarded as an invasive pest species, and state and local governments, and in some cases landowners, have a responsibility to manage these cats, often using lethal methods, such as shooting, poisoning and sometimes blunt-force trauma [20,21,22,23].

Cats in urban and peri-urban areas that are identified as feral based on behaviour and appearance are not required to undergo a minimum holding period in a shelter or animal management facility before being killed [24]. However, research clearly demonstrates that it is not possible to distinguish between feral and domestic cats or their adoptability based on behaviour. Many cats are fearful and stressed in local government animal facilities (municipal pounds) and animal welfare shelters where trapped cats are taken, and appear aggressive or timid, resulting in high kill rates for healthy cats [1,11,25,26,27,28]. Even owned pets can appear fearful and stressed when trapped, resulting in incorrect classification [19,22,25,26,27,29,30,31]. To address this issue, the peak animal welfare organization in Australia, the Royal Society for Prevention to Cruelty to Animals (RSPCA), and some federal [19] and state government documents [22,32] recommend definitions based on how and where cats live. Based on these definitions, domestic cats are fed intentionally or unintentionally by humans, and live in the vicinity of humans. Domestic cats are subcategorized into owned, semi-owned (fed by people who do not perceive ownership), or unowned (receive food unintentionally from humans). In contrast, feral cats live and reproduce in the wild with no support from humans, and survive by hunting and scavenging.

This is an important recommendation since it would mean that cats living in the vicinity of humans, currently deemed feral by state authorities due to a lack of apparent socialization to humans, would instead be deemed domestic and, therefore, subject to different and potentially more humane management methods. In Australia, 3% to 9% of adults report feeding an average of 1.5 cats daily that they do not perceive they own, and are often referred to as cat semi-owners [2,33]. Although most semi-owners feed only one to two cats [2,33], some participate in feeding more, with an average of twelve cats fed in multi-cat situations in Australia [34]. Several cat semi-owners or caregivers may be involved in caring for the same cat group (referred to as cat colonies by authorities), and the care they provide may be organized using feeding rosters [34,35]. Attempts to ban feeding of these cats have had little success, perhaps because, as claimed by some authors, it is difficult to ban compassion, and is costly and difficult to enforce [36].

In other countries, a trap–neuter–return (TNR) approach, whereby free-roaming cats are trapped, neutered, and then returned to the site from which they were captured, is increasingly being used to manage cats in cities and towns, as well as on farms [10,11,37,38,39]. Typically, in TNR programs, kittens and, when possible, friendly adults, are rehomed [34]. When applied with high intensity and purposefully targeted, these programs are documented to reduce cat-related complaints, cat admissions into municipal animal facilities (pounds) and animal welfare shelters and, therefore, decrease the killing of cats. For example, a reduction of 64% in the number of complaints, 32–66% in the number of admissions, and 60–95% in the number of cats killed have been reported over 2 to 3 years [11,37,38,40,41,42].

In Australia, under state and local government legislation relating to biosecurity (feeding feral animals), animal care and protection (abandonment), and domestic animal management (wandering cats), TNR is illegal. It is still practiced on a small scale [34,35], particularly in some states, such as New South Wales and Victoria, where legislation is less stringent, authorities more lenient, or enforcement less robust.

Because management by population control using TNR is limited in application in Australia, authorities typically respond to complaints about free-roaming cats using an enforcement-centred approach, such as trap–adopt/kill or trap–kill [24,43]. This results in large numbers of healthy but fearful, stressed, timid or shy adult cats, and pre-weaned kittens being killed [1]. Moreover, although most trapped cats are humanely killed by lethal injection performed by veterinarians in shelters, municipal pounds, or in private veterinary practices under arrangements with local governments, if cats are deemed feral by local government authorities based on behaviour, they can also be managed by shooting, if this is not considered to pose a risk to the community [22]. The impact on cat welfare of different methods of killing is highly contentious, but beyond the scope of this paper [44]. More relevant here is that no consideration is typically given as to whether the cats in question are being actively supported by human caregivers or not, or even if they are unidentified owned cats [24].

The impact on shelter workers of animal euthanasia (killing) is well documented, with participation in this process being associated with negative psychological effects, including depression, traumatic stress, suicide, and substance abuse [36,45,46,47,48,49,50,51]. In one recent study, physiological indicators of stress in animal carers were elevated during the process of killing, and involvement in making decisions about which animals are killed was found to be predictive of traumatic stress [52]. The complex and poorly understood relationships between occupational stress, traumatic stress, and long-term mental health outcomes in shelter workers who engage in animal killing have resulted in the implementation of various interventions [53]. Given the potential severity of the effects on mental health, it has been proposed that all shelters should engage mental health workers, such as social workers, to mitigate the risks and mobilize protective factors for workers [52,53,54,55].

Although the adverse effects of killing animals on the psychological health of shelter workers is well documented, we could locate no previous research investigating the psychological and social impacts of lethal cat management on the citizens who care for free-roaming cats. However, cat caregivers (semi-owners) have reported being emotionally attached to the cats they are caring for [33], so it is likely that they suffer negative mental health impacts when the cats they are caring for are trapped and/or killed. This is supported by the literature documenting grief and mourning in companion animal guardians and animal care workers, including those who work in animal shelters, veterinary clinics, and wildlife rescues [56,57,58,59]. Anecdotal reports in social media document profound effects on cat caregivers when lethal methods are employed in response to complaints, with caregivers reporting symptoms, such as nightmares, anxiety, insomnia, headaches, and other physical ailments [60,61]. To our knowledge, such anecdotal reports have never been formally investigated.

We believe this to be a significant omission from the literature on the effects of cat control measures. If psychological harm to citizens is formally documented, then local government officials would be wise to consider these adverse effects when deciding on the most appropriate approach for the management of cats in circumstances in which one or more human caregivers intentionally support these cats.

In December 2020, local and national media in Australia reported that a cat cull by shooting had taken place at the Port of Newcastle, a large industrial port in the state of New South Wales, on a breakwall—a permanent barrier constructed at a coastal area that protects a harbour or shore from the full impact of tides, currents, waves, and storm surges. The breakwall and port are under the authority of the Maritime Authority of NSW (state government). The port was privatized in 2014, and the joint Chinese/Australian partners have “obligations to provide safe public access to the breakwalls” under their 98-year lease conditions [62]. Multiple cats were living on the breakwall, being supported by local caregivers. The mostly female caregivers, some of whom belonged to a group called the ‘Stray Cat’s Project’, had been caring for the cats for several years. They indicated that their caregiving was known by authorities for at least five years, and included using a TNR program to reduce numbers from 100 to about 40.

According to a report published by the Australian Broadcasting Commission [63], several cats were maimed or blinded during the cull attempt conducted by a licensed and experienced contractor. The report noted that, after the cull took place, those involved in caring for the cats arrived at the site to discover trails of blood, missing cats, cats with open, gaping wounds, and cats with broken limbs. This is clearly unacceptable from an animal welfare perspective, and hundreds of people subsequently gathered to protest the cull, demanding that the future planned culls be cancelled [64]. This event also provided a unique opportunity to investigate short- to medium-term impacts of this lethal, enforcement-centred approach to urban stray cat management on local cat caregivers. The aim of this study was to gain a better understanding of the motivations of stray cat caregivers, and the relationships between them and the cats they care for. Further, this study aimed to explore how caregivers involved in the caretaking of multiple cats perceived the event of the Stockton Breakwall cat cull, and to explore any potential psychological impact on caregivers.

## 2. Materials and Methods

### 2.1. Research Design

This study used an exploratory approach [65] to enable valid knowledge building about the impacts of stray cat culling on those who care for them. The lived experiences of cat caregivers were at the centre of this research to ensure the voices of marginalized women, who appear to have been neglected in the decision-making process to undertake a cull of cats which they had been caring for, are amplified. The population of this study were caregivers of cats living at the Stockton Breakwall, located at the Port of Newcastle, New South Wales. Semi-structured interviews were conducted by a trained counsellor (VR) and used to explore the thoughts, feelings, and emotions of the caregivers (participants) regarding stray cat culling, and to gain an understanding of any health and psychological impacts experienced because of the cull. This enabled a deep understanding of the lived experiences of these cat caregivers, and the potential impacts on their health and wellbeing.

### 2.2. Participants

The very specific nature of this study required a targeted recruitment process whereby known caregivers of the Stockton Breakwall cats were contacted via social media and invited to participate. Given that there was anecdotal evidence of trauma and distress experienced by caregivers, recruitment and interviews were conducted by a qualified counsellor to mitigate further distress and provide support if necessary. A total of six caregivers, who identified as female and were estimated to be in middle to late adulthood, were recruited for this study between October 2021 and December 2021 (cull occurred December 2020). Two additional caregivers were invited to participate but declined.

### 2.3. Data Collection

Before commencing the study, ethics approval was obtained from the University of Queensland Human Ethics Committee (2021/HE001680). Two forms of sampling were used in this study: purposive and snowballing [66]. Purposive sampling was used to reach potential caregivers involved in caring for the Stockton Breakwall cats at the Port of Newcastle. Specifically, the social media platform Facebook was used to advertise this research and call for voluntary participants. Snowballing strategy [67] was also used, involving using word of mouth to access those not engaged with social media platforms. Interested persons were encouraged to contact a member of the research team (VR) via email. Potential participants were then contacted in return and were provided with a Participant Information Sheet (PIS) and consent form, before scheduling interviews at a mutually agreeable time. The PIS informed individuals that participation was voluntary and confidential, and that no information that could disclose their identity would be published without their consent. Participants were also informed that they did not have to answer any question they felt uncomfortable answering and that they were free to withdraw from the study at any time for any reason. The PIS informed individuals that the interview involved discussing topics that some individuals may find upsetting, and should they require any assistance and emotional support, they could access support and speak to a counsellor. To this end, the names and contact details of three counselling support lines, including the university’s counselling and crisis line, were provided in the PIS.

Semi-structured interviews were conducted via telephone and were voice recorded. Before beginning the interviews, participants were read the PIS and consent form to which their verbal consent was provided. Interviews lasted between 46 and 88 min, with the average length being approximately 65 min. Questions focused on three key issues—the participant’s motivations for caring for the cats, their immediate response to the cull, and any longer term impacts they personally experienced. The interviews occurred approximately 12 months after the culling event.

Once all interviews were completed, they were transcribed by a professional transcription service (Pacific Transcriptions^®^, Brisbane, Australia). The text was analysed independently by one author (RS) using thematic and narrative analysis [68] to identify comments related to the three primary areas of concern and to interpret each participant’s story of the lived experience of the cat culling event, respectively. Extraction was confirmed by a second independent analyst (VR) and interpretation was discussed among the research team.

## 3. Results

The caregivers had been caring for the cats for between 1.5 years and 18 years (average = 6.75 years) (Table 1). The frequency at which the caregivers attended the Breakwall to care for the cats ranged from once per week (*n* = 1) to twice a week (*n* = 1), three times a week (*n* = 1), and 4–5 times a week (*n* = 3). The type of care provided included feeding, supplying fresh drinking water, administering first aid (e.g., removing fishing hooks, fishing lines, and plastic bags from cats), trapping the cats for medical attention and/or neutering, and providing the cats with human interaction and socialization. Feeding rosters were established by the carers to ensure the cats were fed and watered twice daily.

The inductive approach to the analysis resulted in the extraction of several main themes and sub-themes from the interview transcripts. These have been tabulated and context examples are provided (see Table 2). It was also observed that the caregivers commented on the broader social and political impacts of the event. The discussion of these broader themes is beyond the scope of this paper, which is narrowly focused on why the caregivers were motivated to care for the cats and any immediate and long-term individual emotional and psychological consequences of the cat culling event.

### 3.1. Caregivers’ Motivation to Provide Care for the Stockton Breakwall Stray Cat Colony

Caregivers were asked how they became involved in caring for the Stockton Breakwall cats, with their responses falling into two main categories—a general concern about the health and welfare of the cats, and the personal relationships they subsequently developed with individual cats.

### 3.2. Animal Welfare Concerns

Caregivers commonly voiced that their primary concerns were for the welfare and wellbeing (care) of the cats. These concerns motivated them to provide food and water, encouraged them to consider the safety and protection of the cats, and motivated them to decrease the numbers of cats on the wall by facilitating the adoption of kittens and suitable cats, and by neutering (desexing) and returning the cats to the Breakwall who were not suitable for adoption. Not all cats were suitable for adoption due to constraints, such as a lack of resources, limited numbers of suitable homes, and some cats being deemed too timid or shy to be rehomed. Cats living on the Breakwall were deeply cared about and for by the caregivers. The caregivers relayed their thoughts and feelings toward the cats and their desire to help ensure all cats were looked after—that their care, needs, and safety were tended to. Caregivers believed the cats required human intervention to ensure their good welfare as injuries or illnesses were relatively common. The quotes below typify how caregivers viewed the Stockton Breakwall cats and provide examples of the motivational factors driving them to devote their time and care:


*“The number of cats out there—it was concerning because while the cats looked well enough and they were obviously being fed, yeah, they were still quite skinny and that, and I just thought I’ve got to help these cats.”*



*“…they looked like they didn’t have enough to eat. When I first started there was approximately 100 cats out there so you would assume that even if one person walked along, there would always be some that were missed… It was upsetting enough that I thought I actually had to do something… They just were not healthy looking. Obviously, some of them had cat flu and various other issues. Yeah… I wouldn’t have been able to continue walking out there without helping.”*



*“It was just too big a project. I was like, look if we desex one cat a week, one cat a fortnight whatever we can manage, by the end of the year, that’s 26 to 52 cats we’ll have done. You know what I mean? So, if we just chip away at it slowly we should be able to get there, and have them all desexed and all the ones that can be rehomed, rehomed.”*


The caregivers also relayed how the safety of the cats was often at risk due to harmful debris left in the environment, such as fishing lines and plastic bags. There was concern expressed about incidences of intentional harm and injury inflicted upon the cats by members of the public. The caregivers’ motivation to care for the cats beyond simply providing food and water is evidenced in the quotes below:


*“If there were fishhooks in their mouths, we would try and get them out. I’ve even taken antibiotics out there for cats that have had obvious infections.”*



*“If we saw an injured cat then you would obviously try and get it. In fact, I have one here at home who was out there that had—his back leg was swinging. Both bones had been completely snapped in half, so I brought him home—and he’s now my darling cat.”*



*“… then we became aware that not everyone liked the cats—that there were fishermen out there that didn’t like the cats—that there were people out there that were wanting to hurt the cats.*



*“One particular time out near the Adolphe wreck, I stood there for about three-quarters of an hour preventing him from throwing the fishing line into the rocks to damage cats.”*



*“…two different men, one 70 to 80 [years] who had dogs who would ‘sic’ the dogs onto the cats… I’ve stood over the years, in front of where the cats were, to prevent dogs attacking the cats on many occasions, many occasions, but those two men at different times were the worst, because they were doing it deliberately. And occasionally a cat was killed that way.”*


### 3.3. Relationships with Individual Cats

The connections cat caregivers had with individual Stockton Breakwall cats was evident for all the caregivers. They conveyed having bonds and special friendships with the cats using words, such as ‘love’, ‘my cats’, and ‘family’. The level of connection was evident when the caregivers talked of the individual cats by name and pointed out their favourites, when they voiced concern for the wellbeing of cats who ‘went missing’ after the cull, and when they shed tears over the deaths of the cats killed in the cull during the interview process.

Caring for the Stockton Breakwall cats further cemented the deep bond which the caregivers had with the cats. When asked to describe their relationship with the cats, the caregivers relayed having a profound connection with them:


*“They sort of like become your own cats. Even though there was 100, there were still very special ones…”*



*“…the most beautiful pets anyone’s ever had. It says a lot about the label they get. To have these bonds, it’s like having a million children at your feet. We name them all. They all have their names and they’re just so special—so, so special, you know. It is, it’s like having your own child. I have a child, but when they can’t talk and they’re looking at you to keep them safe and fed and the excitement of you being there—because some of them, they just didn’t want to eat. They just wanted to hang out with you, and they’d walk with you. So, I’d just stop and sit down and have a little chat.”*



*“I had a particular cat who is now called Thunder, but he used to come and sit on my lap every morning, and in the winter and when it was raining, I’d open my jacket up and he’d snuggle up. One day, he went missing but I later found out that this other group had him… Please let us know when you catch one, so that we know not to worry that one’s missing.”*



*“I had a favourite called (Nala) and she was one of the ones that got killed… There’s people that really, really had such strong feelings for these animals… They are very loved.”*



*“They’re not feral. They’re pets waiting to go home, they really are. They’ve proven that to all of us that care for them. They just deserve better.”*


### 3.4. Immediate Emotional Impact of the Cull

When provided the opportunity to discuss their immediate response to the cat cull, caregivers described the scene they were met with on the morning after the event using words such as ‘horrific’ and ‘bloodbath’. In response to this event, caregivers described their immediate emotional responses using words such as ‘traumatic’, ‘mortified’, ‘disbelief’, and ‘shock’. Their immediate responses to the cull also included feelings of betrayal. The immediate emotional impact is illustrated in the quotes below:


*“…the worst area. There was blood everywhere. All over the rocks, all over the pathway, like drag marks. So, once I’d sat with them, I’m going, ‘far out!’—something horrendous has happened here… I just started crying because the realization that out of the cats that were there, they probably only spotted about five. It was like, oh my God, what the hell has happened out here?”*



*“… two men came back in sort of like council suits, and they had some wheelie bins with them. They proceeded to scrub the blood away. They had cleaning products, and they were cleaning up the mess. We asked them what they were doing, and they laughed at us. That’s when we knew that this was something way bigger than we ever imagined.”*



*“We looked over onto the rocks… There was this trail of blood. I said, there’s a cat down there—there has to be a cat down there. She just climbed down and sure enough, she pulled out Lily who was the headline of the Breakwall. She’d been shot straight through the head. She’s blind.”*



*“… we were left with very many injured cats and also cats that had got away and passed away within the rocks. So even though we didn’t know who they were exactly, the smell was absolutely horrifying.”*



*“I kept calling out, Charlie, Charlie. Suddenly he pops up with his leg just hanging off him, coming up towards me, and I thought, oh my God. Thank God you’re alive. But his best friend, Max, had died, and here he was all alone, injured, terrified, not sure of what was going to happen next. It was just brutal. It was absolutely brutal.”*


### 3.5. Long-Term Psychological Impacts—The Aftermath

The caregivers reported decreased levels of daily functioning and several negative impacts on their wellbeing following the cat culling event. For some, approximately 12 months after the event, these impacts were still felt. Caregivers also spoke about difficulties related to not knowing the fate of some of the cats and being unable to say goodbye. The following excerpts from the interviews provide more context:


*“… when it happened and I knew I was obviously affected… I took a month-long service leave when it happened from my job, and that was to spend time out there trying to help the cats that were still out there, and also to deal with the emotional side of it, and deal with the rescue side of it.”*



*“… we’ve shed many a tear out there when you find a cat dead or a concern that some are missing. Because so many went missing…”*



*“… it’s just the pressure of everything. I mean I didn’t eat. I couldn’t eat for weeks. I still—I’m 38 kilograms or something. I’m that thin and it’s because when the stress of the cull happened, I literally couldn’t eat.”*



*“I still get emotional and it’s certainly moving on the 12-month mark. Thinking about that is really quite hard for myself and the other feeders, but I feel like I’ve—I don’t think I’ve fully dealt with it… People are okay if I get teary.”*



*“We had Scritch with a broken leg. We had Charlie who had been shot in the leg. We had Maggie who’d been grazed along the neck… To this day, it still impacts me.”*



*“… the thing that stays with us, the cats that they actually picked up and took away in a garbage bin, were they dead? Did they make sure that they were dead? We just would have liked to have had them scanned [for a microchip] so that we know who they actually took away—where did they take them? Not that it matters in one sense, but it does to us because we just wanted to know who they had.”*



*“… we want at least to say goodbye to them… We want their bodies. We want to bury them, or we want to know who’s dead or who’s injured amongst the rocks.”*


Many of the caregivers expressed concern for the long-term welfare and wellbeing of the Stockton Breakwall cats in the wake of the lethal cull. Some described feelings of self-blame as well as fear and trepidation when they return to the Breakwall each day to undertake their caring duties; fearing they may find more cats killed or injured. The long-term psychological impacts on the cat caretakers are expressed in their own words below:


*“… it’s just a constant fear that they will do it again… Just the feeling that we let them down because a lot of the desexed ones… they weren’t tame enough to rehome… We put them back on the wall… Maybe if we hadn’t have let them go back there, they wouldn’t be dead now. But they weren’t tame enough to rehome.”*


Caregivers conveyed feeling betrayed by the Port of Newcastle and that this had significant impact on their ability to cope and process the cat culling event. Specifically, the caregivers felt that they and the work they do was disregarded in the decision to initiate and fund the lethal cull.


*“…there’s still an injured cat out here for God’s sake. I mean, it’s nowhere near ended. So that just annoys the shit out of me, the fact that—I mean everyone makes mistakes but at least own it and try and make up for your mistake—try to right your wrong.”*


Several respondents reported that the cull was initiated after a complaint arising from an incident when a child fell off their bicycle when a cat ran across the Breakwall in front of the bike. The Port of Newcastle website stated they “engaged a licensed and experienced external contractor to help control feral cats on the Breakwall to reduce risks to the community, native fauna, and the environment.” While the original statement has since been removed (originally accessed April 2022), it can be seen documented in local news media posts [60,62]. The cat caregivers expressed care for wildlife as well as the cats, but their observations of the wildlife–cat interactions happening at the Breakwall did not raise concerns over this issue.


*“… lots of people who were concerned about the cats damaging the wildlife but the native rats, the Rakali, well they thrived from the cat food. They intimidated the cats.”*



*“… they should be allowed in that environment because there really isn’t any wildlife to speak of that the cats are a danger to. I have never seen a pile of feathers out there where a cat has caught a bird. Most of the birds there are seabirds such—like seagulls. There’s crows. Well, the crows chase the cats anyway. There’s native water rats out there. But the water rats actually eat the cat food with the cats. The cats don’t seem to attack them. In fact, I’ve seen water rats chase away cats, and bite a cat’s tail so that the cat would leave and he could get the food. So in terms of native wildlife, I don’t see the issue but that is a concern to me.”*



*“They weren’t causing any problems with native wildlife. The rakali that are the native water rats used to cohabitate with them and share their food. They weren’t causing any problem there.”*


As a consequence of feeling betrayed, what was also evident in the caregivers’ discussion was a pervasive distrust of the authorities who organized the event. The quotes below provide some insight into the perceived response of the Port of Newcastle after the cull, and the caregivers’ thoughts and feelings relating to the post-cull assistance from the Port of Newcastle:


*“…even today there’s still one cat there that was shot through the leg—front leg whose leg now can’t bend… Seeing him there like that every day for nearly the last year and trying to catch him to get him the help he needs. The Port never did anything about those injured cats. We caught them all. We’re still trying to catch whose still there. They never did anything. They just don’t care.”*



*“The Port offered us Lifeline (Lifeline is Australia’s leading suicide prevention service. They are a national charity that provides all Australians experiencing a personal crisis with access to 24-h crisis support). They gave us Lifeline’s link. I was like, you’re kidding me. You’ve fucked over one charity—excuse the French—and now you’re going to send us to another charity, when you’re a multimillion-dollar company, to get some help… I was not going to go through that making phone calls when I’m feeling like topping myself. Like not really, but you know what I’m saying. Like needing someone to talk to and then the phone rings out. I’m not even going to go there. Don’t even suggest ringing Lifeline to me, because that would top you over.”*



*“You stood all over us, one Newcastle charity, and now you’re going to use Lifeline Newcastle, another Newcastle charity, to mop up your mess. Get some respect and own what you did. You know what I mean? Instead of—like yeah that really annoyed me so much.”*


Additionally, the caregivers spoke about the physical and behavioural impacts on the remaining cats, on other caregivers, and on the public:


*“… some [cats] are just so scared of people because they’ve been given so much grief since the cull. I regularly experience people throwing rocks at them, trying to kick them, trying to go at them on their bikes.”*



*“… then the extra trauma was finding the wounded cats. Many of them became more furtive because after this experience they were hiding, so that added to not only their pain, but the upset of so many, not just the cat ladies but so many other people.”*



*“… very bittersweet feeling now when I go out on the wall, because it’s wonderful that there’s only so few cats, but the reality is, are we even going to get these ones, and what is their future?”*



*“Just upset, very upset. Not just for the animals, but for the girls involved because I know what a toll it takes…”*



*“There’s a couple of the ladies who aren’t there anymore. It just got too much for them… I can’t leave them (the cats). I can’t leave. I can’t turn my back on them. I’d feel like I’d let them down if I left…They can’t say it was successful in any way, because they left cats there severely injured and left them to die.”*



*“He can’t even go out there and walk in this most beautiful spot in all of Newcastle. He can’t even go out there, because he found that cat and he’s scared; he doesn’t ever want to do that again.”*


## 4. Discussion

In the case study described, lethal enforcement-centred management for the cats being fed daily by cat caregivers was implemented by the Port of Newcastle, NSW, Australia to “help control ‘feral’ cats on the Stockton Breakwall to reduce risks to the community, native fauna, and the environment”. Several respondents reported that the cull was in response to perceived risk to humans, after a child fell off their bicycle when a cat ran across the Breakwall.

The mismanagement of the culling process was evident from the reports of the cats left badly injured, and it would be instructive in the future to debate the relative merits of the various methods of killing cats from an animal welfare perspective. Engaging in this debate is beyond the scope of this study, which focused on the effects of the event on local residents who cared for the cats, often on a daily basis. There are several important findings from this study exploring the lived experiences of the cat caregivers affected by the culling, including the strong bonds the caregivers have with the cats and the short- and long-term impacts on their psychological health. We maintain that these findings should be considered when authorities are considering management methods for urban stray cats.

Several main themes arose from the interviews with the six cat caregivers, whose cat caring experience ranged from 1.5 to 18 years. These themes relate to their motivations to provide care, the immediate emotional impacts of the cat cull, and the long-term consequences of the lethal event.

### 4.1. Motivation to Provide Care for the Stockton Breakwall Stray Cat Colony

The caregivers of our study reported considerable concern regarding the health and safety of the cats, and they also described the lengths to which many of them went to ensure the cats’ good welfare. They reported that the cats on the Breakwall were sometimes afflicted with injuries or harm due to environmental debris and human cruelty. Further, concern was also reported for other animals and the public in relation to the presence of harmful debris, and their efforts to remove it from the Breakwall were described.

Free-roaming or stray cats in cities and towns are frequently fed by people who are compassionate and who enjoy interacting with cats. They feel responsible for improving their health and welfare and commit substantial time and finances to their needs, despite existing legal and financial difficulties [34,35,43]. These people are considered semi-owners, and most feed one to two cats. In some cases, 10 to more than 100 cats are fed, and especially when larger numbers of cats are present, care may be provided by multiple people and organized through rosters [34,35]. The respondents in our study clearly identify themselves as belonging to this broader group of cat semi-owners.

Concern for the welfare of urban stray cats is often centred around a person’s love of animals, sympathy towards cats that may be hungry, injured, or unhealthy, and ethical concerns [69]. Caregivers often provide not only food and water, but also first aid and (self-funded) veterinary attention for cats within their care, including neutering [34,39], as did the caregivers in our study. The caregivers in our study expressed a desire to reduce the number of cats by neutering and adoption, out of concern for the cats’ welfare. Indeed, the caregivers in the current study reported that through neutering and adopting socialized cats and kittens, they had reduced the population from approximately 100 cats to 40 cats. This is consistent with the reasons cited by respondents for beginning TNR in an Australian study: it was a humane (compassionate) approach to cat management and, even if illegal, an effective way to reduce the cat population over time [43].

### 4.2. Caregivers’ Bonds with the Cats

The motivations for care were further strengthened by the bonding and relationships each participant felt with the Breakwall cats. This study revealed the strength of the relationship between caregivers and individual cats, even though they reported there were 40 or more cats at times, and they did not own them in a legal sense. They nonetheless felt responsible for their welfare. The caregivers described their bonds with the cats as being as strong as the bonds with their own pets and asserted that they thought of them as their cats. They even described them as being like their own children, in that the cats looked to them (the caregivers) to keep them safe and fed. They “all had their names and personalities”. This relationship appeared reciprocal, evidenced by the close interactions described by the caregivers between themselves and individual cats; the cats would curl up in the caregivers’ jackets, butt them for head scratches, and run to meet them on the Breakwall. The benefits of human–animal relationships for psychological and psychophysiological health in people have been well established in the literature [70]. This is supported by a study of cat caregivers in Australia in which the caregivers reported that feeding cats “makes me feel good”, “it is the right thing to do”, and “the people who I care about would approve” [33]. Our study provides further evidence of the positive impacts of human–animal interactions and relationships, but unfortunately also highlights the psychological trauma that can result when the relationship is unexpectedly severed.

### 4.3. Psychological Impact of the Event

During the interviews, the caregivers described the culling event as ‘horrific’ and ‘traumatic’. Since the caregivers were not informed the cull was to occur, they had no opportunity to prepare for the event, so it is perhaps not surprising that the caregivers also described the event as ‘shocking’. The ‘bloodbath’ that they witnessed may have also intensified their feelings of shock and horror. Events that are unexpected and out of an individual’s control can have the potential to cause greater psychological impact [71].

The emotional costs of cat management have been documented in shelter staff tasked with killing cats and kittens. Traumatic stress and increased suicide risk have been reported in shelter and animal control staff associated with the euthanasia of healthy animals [47,50,52,54,72]. Grief reactions have also been documented in animal caregivers [56,57,58,59]. The findings extend this research and show that lethal cat management can lead to intense immediate emotional reactions as well as longer term psychological impacts in cat caregivers.

All the caregivers described psychological impacts after the cull, with the impact still being felt nearly one year later. The interviews revealed that the cull affected their daily functioning, with one participant reporting that they took time off work and other caregivers reporting persistent weight loss and nightmares after the event. Nightmares may be characterized as an intrusive symptom which, together with initial feelings of horror and persistent negative changes in mood, may be indicative of posttraumatic stress [73]. This is consistent with findings that animal rescue workers exposed to euthanasia are more likely to be psychologically impacted than those who are not exposed to euthanasia [74]. What is different here though is that this was a mass killing conducted by shooting, rather than what might occur in an animal shelter environment where animals are likely to be killed individually via lethal injection. To this end, it is possible that the former may have an even greater psychological impact. Most urban stray cats that are managed by enforcement are trapped, rather than killed outright. Killing the cats being cared for after trapping them may on the other hand have similar traumatic impacts if caregivers are not informed or if they disagree with the practice, and/or the fate of the trapped cats remains unknown. This is an area which requires further research.

Strong bonds with the cats were evident in these caregivers. Therefore, not only did they experience the event as traumatic, but they may also have experienced grief from the loss of individual cats [59]. In interviews, feelings of self-blame are evident in relation to the returning desexed cats, who could not yet be rehomed, to the Breakwall, even though the caregivers could not have foreseen the fate of those that were returned. These feelings of guilt and self-blame can commonly manifest in those who are grieving and can have a detrimental impact on later adjustment [75]. A review found feelings of guilt can negatively impact adjustment in those who are bereaved, with studies finding guilt is associated with outcomes such as traumatic reactions, impaired physical health, and psychological distress [75].

The intensity of a grief response can be a function of one’s level of attachment, whereby those more closely attached may experience grief more intensely than those less attached [59]. Caregivers in our study referred to the cats as ‘pets’ and ‘children’, so it is likely that the grief experienced from this traumatic, sudden, and unexpected loss was profound. The lack of closure resulting from not knowing what happened to the ‘missing cats’ could have compounded these feelings of grief. This form of loss, known as ambiguous loss, has been linked with long-lasting, detrimental impacts on individuals [59].

Not only is there evidence of posttraumatic stress and grief resulting from this event, but there is evidence of feelings of betrayal and altered perceptions of authorities. An implicit social contract between the cat caregivers and the authorities was potentially violated, which may have contributed to the event being difficult to process for some. As one interviewee noted: “They didn’t even tell us. That’s, I think, the hardest part, knowing that all these years we’d had this good relationship with the Port of Newcastle, they at the end did not honour or respect us as people who really cared for these animals. That’s a hard thing to process.” This feeling of betrayal may have also intensified the impact of this event and perhaps led to long-term distrust.

### 4.4. Implications and Considerations Arising from the Stockton Breakwall Cat Cull

Although the authorities at the Port of Newcastle deemed the cats feral, this was not consistent with how the caregivers viewed the cats—“they’re not feral. They’re pets waiting to go home”. Nor was it consistent with the RSPCA definition of feral cats in their Best Practice Domestic Cat Management report (RSPCA 2018) or the Australian Federal government’s Threat Abatement Plan for Predation by Feral Cats, adopted in 2015 [19]. In these documents, feral cats are defined as those which are unowned, unsocialized, have no relationship with or dependence on humans, survive by hunting or scavenging, and live and reproduce in the wild. In contrast, domestic cats are defined as cats with some dependence (direct or indirect) on humans. Despite these definitions, local government and animal management officers often determine a cat is feral based on behaviour and appearance, which allows the cat to be killed immediately after being trapped [24,25,26,27,28] or through shooting when it is not considered a risk to humans or pets. We believe this is inappropriate if cats are living in the vicinity of humans. On the evidence presented by the caregivers in our study, the cats in their care were being fed regularly and the majority were well habituated to people. Therefore, the Stockton Breakwall cats were not feral and should not have been (mis)managed in this way.

The Port of Newcastle’s aim was “to reduce risks to the community” but the severity of the adverse psychological impacts, and the morbidity rate amongst the cat caregivers we interviewed, was far greater than would be expected as a risk to the community if the cats had remained at the site. We therefore suggest that potential legal ramifications should be considered before authorities intentionally choose a method of management that is likely to inflict substantial harm on community members.

Given the reported ramifications of the lethal cat cull to both the caregivers and the remaining cats on the Breakwall, it would be prudent to make mention of the alternatives that could have been employed to address the presence of the cats on the Breakwall. Specifically, this group of cat caregivers was, according to the caregivers, making a significant impact on the cat numbers, and had reduced them from over 100 to less than 40 over a 3-year period. This was achieved by some caregiver’s using TNR, which typically consists of providing food, some veterinary care as required, neutering the cats to reduce their numbers over time, and adopting kittens, and when possible, social adults [34,35,41]. Although technically illegal in Australia, TNR has been implemented successfully when supported by authorities, often after traditional methods have failed to reduce complaints or cat numbers [2,35,43]. This method could also be considered a care-centred approach to cat management, as it protects against psychological and emotional trauma in those who care for, and are deeply attached to, the cats. Further, the care-centred approach has been shown to be successful in multi-cat situations in Australia and overseas [34,76]. It can improve animal welfare, reduce the numbers of cats present over time, and reduce complaints from the community [10,11,13,35,37,39]. A care-centred approach to urban cat management is also consistent with the One Welfare philosophy, which aims to balance and optimize the wellbeing of animals, people, and their physical and social environment [77,78]. The benefits of TNR are several: healthy, adoptable cats are provided with forever homes; healthy cats which cannot be adopted are neutered and thus rendered unable to reproduce but are cared for and allowed to live out their lives at their home; and caregivers are afforded the physical and psychological benefits of maintaining a bond and mutually beneficial relationship with the cats [79]. Benefits may not occur if an insufficient number of cats are sterilized to prevent population growth and when best practice is not followed to resolve complaints [34]. Moving forward, the benefits to authorities in adopting a care-centred approach to addressing cat populations that are under the care of people will strengthen community trust and acceptance, as well as contribute to their social license to operate.

## 5. Conclusions

This study demonstrated that lethal enforcement-centred management can be detrimental to cat caregivers’ psychological health, quality of life, and physical health. This is in addition to the clearly unacceptable impact of this approach on the welfare of the cats in question, at least some of which were left with severe trauma and horrific physical injuries. The results provide evidence of the strength of the relationships that form between caregivers and the cats they care for, and the negative impact on mental health and quality of life associated with the implementation of lethal cat management by authorities. Based on these research findings, there may be legal implications if authorities in the future disregard the potential for creating profound adverse psychological damage to caregivers of stray cats, and knowingly implement management strategies which will be harmful to human health and cat welfare. While cats cannot seek legal redress for harms inflicted on them by poor policies, impacted humans are able to challenge the legitimacy of management practices, as was evidenced by the substantial grass-roots protests that followed the poorly executed cull described in this paper.

It is hoped that this research will inform local government and welfare agencies of the negative impacts of current practices and provide evidence that will lead to the adoption of either a care-centred approach with regards to the cat caregivers, or perhaps more generally an approach centred on care for both humans and the non-human animals we feel obliged to ‘manage’. Cat conflicts with other free-roaming animals or people may need to be managed, but this process should be informed by widespread community consultation and compassion. This is likely to provide long-term solutions which benefit the greater community. In a care-centred approach, authorities could assist caregivers to get cats neutered and adopted when possible, as well as assist with the provision of feeding and shelter stations to optimize cats’ welfare and minimize the risk of complaints.

Legislative amendments need to be prioritized to facilitate this change, including clearly defining domestic cats as those that live in the vicinity of humans, and are provided food or other care intentionally, or in some cases unintentionally, and feral cats defined on how and where they live, and not based on behaviour or appearance. Legislative changes would enable a care-centred approach to be implemented with the aim of resolving concerns related to complaints, humanely reducing cat numbers through neutering, and when feasible, adoption, and improving overall welfare.

As one of the caregivers concluded “…it was really heartbreaking, because a lot of cats that died in the cull were just waiting for a home, you know. That’s the really hard part that I personally struggle with, is that so many of them just could not have been there, but they were, and they died. What can you do? I mean, it’s happened now, it’s not going to change, but what we can do is try and advocate for them, for that not to be the way that they die”.

## Figures and Tables

**Table 1 animals-13-00271-t001:** Participant demographics; Stockton Breakwall Cat Caretakers.

Participant	Years of Caring	No. of Days Attending Breakwall per Week
1	1.5	once
2	2	twice
3	3	3 times
4	6	4–5 times
5	10	4–5 times
6	18	4–5 times

**Table 2 animals-13-00271-t002:** Major themes and sub-themes with context examples from interview transcripts.

Theme	Sub-Themes	Context Examples
Caregivers’ Motivation to Provide Care for Cats	Animal welfare concerns Relationships with individual cats	“We had so many cats and it was this real desperation to get them off the wall, to reduce the population… I just thought, this is going to be my way of helping a problem that’s been created by us, by people. I just really wanted to see these cats taken care of, and be part of a solution, not the problem” “We just wanted to reduce the numbers, so there was less cats out there for the reason that we were—you know, there was the people out there that used to call them the feral cats, and say they’d be better off dead or they’d say, well, we want to kill these cats or we want to hurt these cats.” “…the amount of fishing line and dog poo because people walk along the Breakwall with their dog, and they shit everywhere. No one says a thing about that. We’re constantly picking up fishing line. There’s three or four times we’ve actually rescued seabirds that we’ve found in distress with lines around them and hooks…” “One of the cats had a hook in its paw and I realized then the risks to the cats…careless fishing folk, but also lots of people walked out there in thongs and they walked out there barefooted. So that led me to picking up fishing line and so on as well” “…they’re basically the same as a pet cat that you’d have at home. They have names. They have personalities. They have their little traits that they each individually have… The bonds that we have with them are just as strong as the bonds as my own cats that live in my house…we think of them as our cats.” “(Dusty) has been here before. She’s an old soul. I think I knew Dusty in another life. I don’t know who we both were, but I believe we both knew each other before.” “…I always felt like I needed a purpose in life, but I never really felt like I found it till I found the Breakwall cats. I feel it’s my one time in my life that I’ve made a difference and, yeah, I help save cats’ lives now…”
Immediate Emotional Impact of the Cull	Trauma Disbelief Shock Grief	“…we just went down there to feed them like normal and were met with a—just bloodbath of blood everywhere…lines of blood and then they just end at the end of the Breakwall… The whole thing was just horrific.” “…imagine coming home to your own house and finding your cats shot and injured and bleeding and terrified. Imagine coming home to that scene. Well, that’s what we (experienced)—that’s what happened. I think, yeah, the whole thing was just horrific.” “…we just didn’t know what had happened and we didn’t know how many had been killed, and were they killed outright? We don’t even know the ones they took away if they were actually dead. We don’t know what they did with them. We don’t know who they took. We don’t know who died days after…” “…we just felt absolutely grief stricken. I cried like I’d lost all of my pets my whole life a million times over, because I didn’t know exactly who had gone, who was left injured.” “…it was the way in which it was done and the blood that was just left everywhere. There were some attempts to do something with it, but for anyone to go out there, it would have been just—and it was for many locals, many people—so traumatic. There were lots of people traumatized by what had happened.”
Long-Term Psychological Impacts—“The Aftermath”	Complicated grief Betrayal Pervasive distrust PTSD-like symptoms	“Horrific. Months and months and still today of horrific nightmares. Nightmares about cats being injured and jumping into the water and me trying to get in the water to find them and I can’t. Just that repeated nightmare because I couldn’t help them, and I was—felt so helpless.” “They didn’t even tell us. That’s, I think, the hardest part was knowing that all these years we’d had this good relationship with the Port of Newcastle, they at the end did not honour or respect us as people who really cared for these animals. That’s a hard thing to process, that betrayal and being deceived, and just trying to find forgiveness for these people. It was just horrible. It was really horrible. A horrible thing to do.” “That girl will kill herself over the cats. Then to have someone do what the Port of Newcastle did, it’s a personal attack… the amount that had been invested emotionally, personally, financially and the—what’s the word? The attitude, just the attitude of the Port who couldn’t care less.” “I thought they’ve [Port of Newcastle] got blood on their hands from the cats, now they don’t want blood on their hands with a human life as well. They thought maybe that oh gee, someone might be so upset they might kill themselves, and gee that would make us look bad, wouldn’t it?”

## Data Availability

Most relevant data are reproduced in the text.

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
