# Peer review of "The Impact of Lethal, Enforcement-Centred Cat Management on Human Wellbeing: Exploring Lived Experiences of Cat Carers Affected by Cat Culling at the Port of Newcastle"

_animals, 2023, doi:10.3390/ani13020271_

Round 1
Reviewer 1 Report
This is a very interesting new addition to our knowledge about free-roaming cat caregivers. Most of my comments are about terminology and a few places to make the arguments more complete or clear. See below for specific comments. I did find the manuscript to be a bit wordy in the introduction and discussion. Please closely review the goals of the manuscript and each paragraph and see if some tightening of the text is possible.
The simple summary and abstract use free-roaming or free-living and stray and feral terms. Please use free-roaming for all outdoor cats, and unowned free-roaming for that subset. I think it would also be helpful to state what the definition of feral is in Australia for clarity as there are many definitions in use. So, line 33-34 would be “free-roaming cats” or “presumably unowned free-roaming cats”. Please try to keep this simple and clear. Alternatively, use more generally clear terms throughout except when needed to explain the law and terminology in Australia. Line 41: without the Australian definition of feral this doesn’t make sense. And if domestic cats are owned cats, that should be the terminology and definition earlier in these sections. It isn’t till lines 83-4 that we learn how feral is interpreted in Australia and for this manuscript. I think something about this, perhaps using non-legal Australian language would help set the stage and make the conclusions clearer. By proposed definition then, any fed cats aren’t feral—that may be all that is needed in these sections.
Line 28-29 doesn’t describe mutually exclusive categories as written. Since semi-owned cats could be lost cats or offspring of free-roaming unowned cats. Please clarify here and in the introduction. But see below about the discussion and use of this term.
Line 93: “colonies” sometimes has negative connotations of many poorly cared for cats. I’ve been using cat groups as an unencumbered term for this.
Line 97: instead of stray I would use unowned free-roaming cats. That is clear and without too many confusing national or legal definitions.
Line 100: if there is a reference please add.
Line 103: please reference these statements or link this sentence more clearly to the following sentence and those references.
Line 121: or even if these cats are owned cats allowed to wander who may present as feral behaving? Reference 18 can be used effectively to demonstrate that even pets who are well socialized to humans can appear feral in animal shelters or unfamiliar environments.
193-4: did the participants coordinate care so someone was there every day? Please clarify in the text.
Line 516-7: I think that the use of the term semi-owners is potentially problematic given the legislative definition of ownership (and its variability from location to location). If this concept is important for this manuscript, it should be more clearly stated as defined by the authors and others, and not a legal definition. Why not call them caregivers throughout?
Line 538: “thought of them as their cats”. Yet many times, anecdotally, caregivers state they don’t own the cats. And the legal definition of ownership may or may not lead the caregivers to be legally identified as owners. Better clarity about the relationship with the cats and whether they consider themselves owners or caregivers with similar relationships is needed here especially. Also, throughout the manuscript when the bond or term semi-owner is used to more clearly make the point that these are close relationships which may or may not be the same as legal or self-identification as owners of the cats.
Throughout the discussion, there are different forms of trauma and emotion discussed. One theme I didn’t see was related to the betrayal: the not-knowing what happened to the cats. Plus, if shooting is to be the method of control, leaving injured cats alive is unacceptable.
Additionally, given the concerns of the caregivers and injuries to cats living in this location as well as the statement that these are pets waiting to go home, I’m a bit surprised there were still so many cats. That juxtaposition should be addressed in the discussion in more detail as it weakens the argument about the attachment to and protection of the cats by caregivers.
One final point to include is that a cull by shooting isn’t going to control the population unless it is repeated over time. If reducing risks to the community was the goal, was there a long term goal of fewer cats or was this a one time complaint driven activity with no real impact on the community (regardless of the harm to the individual caregivers). Seems like some long term plan or data driven approach would also decrease psychological harm.
Reviewer 2 Report
I appreciated the opportunity to review this paper focused on the experiences of community (aka feral) cats and their caregivers in the wake of a cull at the Port of Newcastle in Australia. As the authors note early in the paper, there is a paucity of research on caregiving for community cats, and on the psychological impacts of culling on caregivers. In Australia and other contexts where killing feral cats is legal and Trap-Neuter-Return is illegal, the findings stand to have important implications for guiding policy. To maximize how impactful the paper is, I have several recommendations for revision.
I think what is most jarring about the paper as currently written is how the qualitative data is presented. This is most apparent in the Results section, in which the authors identify themes and then include multiple quotations from participants that capture each theme, but without any of the authors’ analysis. Here, the authors should add their own analysis (some of which already appears in the Discussion section and could simply be relocated and wordsmithed), rather than leave the reader to wade through quote after quote. I also wasn’t sure why some quotations were placed into boxes. It’s not necessary for the authors to provide every example of a theme; in fact, it may be more powerful to trace one or two respondent’s voice on a theme with greater analysis. Given the very small size of the sample (which is fine, given the circumstances of the research and that it would be functionally impossible to develop a large sample of cat caregivers whose cats are culled), I would suggest attaching a pseudonym to each caregiver, and providing some basic information about them (gender, age, race) so that they come alive more as people. Tracing the experiences of named individuals would make the presentation of their experiences much more compelling and engaging, and will make the caregivers more relatable (something that may need to be accomplished as, unfortunately, some people dismiss cat caregivers as “crazy cat ladies”). Providing pseudonyms (or actual names, if the consent process allowed for that) would also enable the reader to see how much the authors are relying on one or two informants, versus the full range of informants. I found the division between the Results and Discussion quite artificial—in a qualitative paper, these would typically be woven together, and I encourage the authors to bring the analysis into their presentation of the data.
While surveying the expansive literature on grief and traumatic loss generally, or on pet loss more specifically, is likely not reasonable for the authors at this stage, the presentation of findings related to the after-effects of the cull on humans would be strengthened by referring to this literature—maybe just adding a few key citations so that readers understand that these after-effects are responses to loss and trauma. There are also ready linkages the authors could make here to the literature on the psychological effects of working in kill shelters for staff and volunteers (see, for example, Frommer and Arluke 1999, Goldberg, DiGiacamo and Reeve 2007, one of the chapters in Guenther 2020, Bennett and Rolf 2005, Arluke 2002). The politics of who is grievable is itself an issue here (even if the authors aren’t addressing it explicitly), and while I think the authors don’t have to go into that in this particular paper, establishing that what these caregivers experienced is a traumatic loss will make their arguments about the cruelty of such culling more evident, and also place the paper into a broader conversation about animal death.
While the sample size that the authors had available is small, their findings are quite powerful, particularly regarding the traumatic consequences of the cull for humans. I would like to see them develop even more strongly worded recommendations in their conclusion. I was puzzled by the suggestion of a carer-centered approach, as opposed to a care-centered approach, especially given the grievous physical and psychological injuries of cats the study documents. In the conclusion, the cats themselves largely disappear, even though we previously learned of their suffering. Bringing the cats back in here—reminding the reader of the brutality of killing cats (and also addressing that “But what if we catch them and humanely euthanize them?” question some readers are sure to ask) and that cat conflicts with other free roaming animals and/or with humans should be carefully evaluated with community participation prior to any program of cat management—would give the paper extra punch at this important juncture of recommendations. And I think advocating for a care-centered approach would decenter humans and recognize and include animals in ways that challenge the idea that humans are simply at liberty to dispose of “nuisance” animals however they see fit.
Less thematic additional suggestions:
· I had assumed the Port of Newcastle was a government entity, but then one of the respondents refers to it as a corporation. What is the Port of Newcastle? If it is a corporation, do policy recommendations need to include legislation that impacts private property owners as well as government agencies? This seems to me an important detail.
· A qualitative paper would not normally have a Results and Discussion section, but rather more descriptive headings.
· Since much of the readership is likely to be outside of Australia, I would suggest the authors include even just a few sentences about the development of Australian views and policies vis-à-vis free roaming cats.
· In the abstract and elsewhere, the authors refer to free-roaming cats as posing a “management challenge” to authorities. It’s my experience that in many parts of the world (including the United States, South America, and many parts of Asia), these cats are actually just ignored—the state functionally abandons them and considers them outside of the purview of animal control. The authors should either add citations about evidence of this management challenge or reword (and I suspect the latter is a better option). I would also encourage the authors to think about why Australian authorities construct these animals as “problems,” when they are not seen as such in so many other places (see also the previous comment).
I wish the authors the best with their revision and hope that this paper will be published after revision.
Reviewer 3 Report
Interesting article
in the methodology I would like to see some statement about how human subjects are protected. For example how identities are kept confidential etc. also given that you failed evidence of psychological trauma in the human subject, what will you do if and when they need more help. You also mention the potential risk of suicide as a result of experiencing this trauma, so how are the human subjects protected here?
The presentation of your findings feels like it could be presented in a cleaner fashion. I don’t understand the purpose of the boxes around some of the quotes. Some of the headers that identify thieves are in italics blending in with the quoted material. I just feel there could be a better way to present the themes.perhaps in addition to changing the header font, you could list the themes together before presenting the support for each team I’m not sure how to help make the presentation clearer. I just identify that for me it got confusing.
Reviewer 4 Report
I would like to thank the authors for submitting their work for review. Unfortunately, I have to recommend the manuscript be rejected as this qualitative work is highly biased and not appropriate for publication in a scientific journal. I list my major concerns below.
1. The authors do not cite primary research literature in their introduction. A paper titles 'Outcomes for cats entering RSPCA shelters' is not an appropriate citation for supporting a list of negative environmental outcomes caused by cats. This tendency to cite inappropriate literature occurs through the manuscript.
2. The authors fail to cite relevant literature such as the state and national legislations they are commenting upon. This article may be read by an international audience that is not familiar with the legislative framework of Australia and hence citations to these bills should be provided. The authors should also provide some specific details within this bills, such that owners of domestics cats are required to have their cats microchipped, sterilised, and wear a collar by 6 months of age so that they can be identified as a domestic cat. This information is particularly important when the authors claim that local government authorities are poisoning cats as on P3L118. The APVMA does not have a poison legally registered for use on cats.
3. The authors use inaccurate language. For example, P3L99 "all cats in towns and cities..." The word "all" is extremely inaccurate. The vast majority of cats are not managed via any method. Similarly, P3L100, "Typically, ... cats are rehomed". Please provide quantitative evidence that the majority of TNR programs involve rehoming animals, otherwise moderate your language. On P4L179 the authors expect authorities to expect that women conducting an illegal activity (and hence unlikely known to authority personnel) to be attached to the subject of their illegal behaviour. A scientific journal is not a place for an emotive opinion piece.
4. The manuscript is not appropriately structured for a journal article. Results should be in the results section, not the second paragraphof the methods section.
5. Authors need to accurately depict the details of their research project in the introduction, rather than just repeat the standard phrases of the global feral cat debate. P10L395, has a very interesting quote from one of the study participants that implies that the Port of Newcastle had agreed to the activities of these women, that they were a known group. Not just a bunch of women hiding their illegal behaviour. This information needs to be conveyed in the introduction! You need to set the entire scene for the reader. The reader cannot hope to understand the psychological framework of this event if we don't have this basic information at the beginning of the article.
6. The disucssion is a repeat of the introduction rather than a discussion of new insights and study strengths and weakness and alternative explanations for the results.
7. The authors need to discuss the interpretations of their research in the context of broader research on animal welfare and management. I've highlighted on P14L172 that shooting is also a stupid and dangerous method to use on a marine breakwall frequented by the public! The probability of richochet and unethically aimed shots is extremely high! There is no way I would ever be given permission by my workplace to manage these cats using a firearm! Our rule book on what type of firearm, ammunition, species, and distance of shot is allowed is over 3 inches thick! I recommend this article briefly discuss the welfare of animal management broadly. There are hundreds of papers on the ethics and welfare of different management techniques that could have been employed and these alternative scenarios should be discussed briefly because much of the horror experienced by the caregivers is probably because this entire situation was badly managed. Replace the long repetitive paragraphs on whether feral cats are good or bad, and instead focus on the management options, the legislation, and the stakeholders involved in these situations.
8. Given the psychological approach to this research the discussion should definitely address the impact of guilt felt by the caregivers as evidenced on P10L387. The caregivers chose to return the cats to the Breakwall exposing them to the danger of lethal management and this decision is evidently affecting some of their mental state.
I have attached a pdf with specific comments.
